# Multiscale Observation Product (MOP) for Temporal Flood Inundation Mapping of the 2015 Dallas Texas Flood

**Elena Sava** [1,*,†], **Guido Cervone** [2,†] **and Alfred Kalyanapu** [3,†]

1　Geospatial Research Laboratory, Engineer Research and Development Center, 7701 Telegraph Rd., Alexandria, VA 22315, USA
2　Geoinformatics and Earth Observation Laboratory, Department of Geography and Institute for CyberScience, The Pennsylvania State University, University Park, PA 16802, USA
3　Civil & Environmental Engineering, Tennessee Technological University, Cookeville, TN 38505, USA
*　Correspondence: esava0107@gmail.com
†　These authors contributed equally to this work.

**Abstract:** This paper presents a new data fusion multiscale observation product (MOP) for flood emergencies. The MOP was created by integrating multiple sources of contributed open-source data with traditional spaceborne remote sensing imagery in order to provide a sequence of high spatial and temporal resolution flood inundation maps. The study focuses on the 2015 Memorial Day floods that caused up to USD 61 million dollars of damage. The Hydraulic Engineering Center River Analysis System (HEC-RAS) model was used to simulate water surfaces for the northern part of the Trinity River in Dallas, using reservoir surcharge releases and topographic data provided by the U.S. Army Corps of Engineers. A measure of fit assessment is performed on the MOP flood maps with the HEC-RAS simulated flood inundation output to quantify spatial differences. Estimating possible flood inundation using individual datasets that vary spatially and temporally allow to gain an understanding of how much each observational dataset contributes to the overall water estimation. Results show that water surfaces estimated by MOP are comparable with the simulated output for the duration of the flood event. Additionally, contributed data, such as Civil Air Patrol, although they may be geographically sparse, become an important data source when fused with other observation data.

**Keywords:** data fusion; remote sensing; damage assessment; hydraulic models; flood; HEC-RAS

## 1. Introduction

The increasing trend of population living in urban areas, along with deteriorating infrastructure and projected hydroclimatic changes, escalates the likelihood of society becoming more vulnerable to natural disasters. The growth in frequency and magnitude of hydrometeorological events can result in loss of life and property, and can negatively affect the economy of a region [1,2]. This is particularly evident during flood events, where human factors can exacerbate the damage caused by a natural phenomenon. One of the key components in preventing and reducing losses is the ability to accurately monitor and model flood extents as well as to provide reliable information about the risks associated with flooding and actionable warning. Accurate predictions often involve a range of interactive activities that become critical safety tools, assisting in providing actionable input to the response community and informing the general public [3].

Assessing the degree of a flood extent with a high spatial and temporal resolution as an event is progressing is a difficult task. Currently, there is a lack of tools that allow professionals to measure flood extents and to make high-resolution assessments in a timely manner. Specialists often rely on river gauge station readings, numerical models, or for surveyors to take manual in situ measurements [4]. However, during a flood event, these resources could be compromised or take time to carry out, resulting in temporal gaps

in data. Additionally, efforts in maintaining gauge stations are often costly and sparse, making it difficult to obtain accurate measurements. For these reasons, satellite and aerial data are often used because they provide multispectral, high-resolution spatial data that can be used to make assessments before, during, and after a flood event.

Remote sensing provides a synoptic overview of the Earth and gives valuable environmental data for a broad variety of scales, ranging from entire continents down to urban regions and spatial pixel resolution ranging from kilometers to centimeters [5]. Despite the many advantages, several drawbacks prevent remote sensing platforms from being solely used during emergencies. Sensor limitations for optical remote sensing platforms are often a serious drawback since no single sensor offers the optimal spectral, spatial, and temporal resolution at the same time [6]. Specific data may not be collected at the time and space most direly required and/or may contain gaps as a result of the satellite revisit time, atmospheric opacity, or other obstructions. Additionally, these platforms are heavily limited during periods of clear sky and daylight conditions. Likewise, airborne platforms are weather-dependent, often require multiple flights at high altitudes to view large flood areas, and only provide information in the visible part of the electromagnetic spectrum (EM), making it difficult to extract information quickly [7].

As one of the ways to overcome the limitations of optical remote sensing platforms, the research community has shifted towards active microwave remote sensing. Due to its all-weather, day and night capabilities, synthetic aperture radar (SAR) data have been used to identify water bodies and estimate flood extent for multiple environmental hazards [8]. Yet, several challenges still occur due to the low-backscatter properties of water with other environmental variables. For example, land cover variables such as vegetation and urban areas cause multiple reflections, increasing the amount of backscatter information being sent back to the SAR sensor. This effect makes these environmental features notably similar to the properties of open water, making it difficult to distinguish between flooded areas and non-flooded regions. Optical and active data have substantially been investigated at various levels of spatial resolutions and accuracy assessments, resulting in visual interpretation often being found to be the most accurate assessment [9].

In occurrences where observation data are lacking or not available enough to provide a complete flood extent, areas at risk are delineated using hydraulic models. Hydraulic models are designed to simulate different flood scenarios which aid in identifying flood-prone areas, and estimating water depth and probable flood inundation [10]. One characteristic of hydraulic models is that they replicate flow based on the topography of the channel and floodplain by applying physical laws to fluid motion with varying degrees of complexity and minimal parameterization requirements [3]. Traditionally, numerical models rely on water level measurements or flow from stream gauging stations to simulate inundation. With the development of high-resolution topographic mapping through the advancement of remote sensing data, models expanded to incorporate structured and unstructured grid meshes for 2D modeling. Digital elevation models (DEMs) often serve as the terrain parameter that is essential to estimate the topography and geometry of a river channel, flood basin extraction, discharge, ditches, etc. [11,12]. Studies comparing 1D and 2D models have shown that 2D models have the capability of sufficiently estimating variables such as flow velocity, inundation extent, and water level depth, all of which are crucial factors for flood risk management. However, alternative studies have also shown significant differences between 2D model performances, specifically in urban environments when estimating velocity. This is due to the presence of small-scale features such as bridges, roads, and buildings, which tend to alter the flow pattern, making it challenging to apply 2D models in urban areas in comparison to rural regions [13]. Because of this model limitation present in urban regions, a detailed validation dataset is needed [14].

*Remote Sensing and Data Fusion for Flood Assessment*

A significant amount of research has focused on the use of remote sensing data for rapid flood inundation mapping from platforms with different spatial resolutions.

Refs. [15,16] used 8-day composite images from Moderate Resolution Imaging Spectrora-diometer (MODIS) to identify the spatiotemporal extent of annual flooding over Cambodia, the Vietnamese Mekong Delta, and Bangladesh. Inundation extents obtained from MODIS are compared with RADARSAT-derived inundation maps and show that it is possible to observe flood dynamics by examining patterns of flood inundation and recessions using coarser resolution imagery, as carried out similarly using moderate resolution data. Similarly, Ref. [17] used 30 m Landsat imagery to measure inundation changes over the Macquarie Marshes located in central–eastern Australia for a 28-year time period. These studies show examples of how the spatial analysis of long-term imagery is a powerful way to measure changes in inundation over large spatial scales. Yet, these types of analyses would be difficult to perform over small regions, urban environments, and shorter time scales because of the spatial and temporal resolutions of the satellite data.

Higher spatial and temporal resolution data are required for an up-to-date inundation over a single event. For example, Refs. [18,19] used multitemporal series data from COSMO-SkyMed to assess flood evolution mapping and damage after the 2009 floods in Northern Italy and the 2011 Tohoku tsunami in Japan. Ref. [18] applied an automated image segmentation algorithm and an electromagnetic surface scattering model to derive flood progression of three consecutive days and recession for post-event analysis. Ref. [19] applied change detection to identify flooded areas and debris over urban areas in the harbor of Sendai.

The availability of new and diverse data sources, paired with advancements in data-driven methods, have transformed the practice of flood model initialization, calibration, and validation. Until recently, only sparse streamflow measurements, available at few locations, were used to spatially validate hydraulic models. Because models can simulate large and diverse two-dimensional domains, the comparisons of simulated and observed measurements at these few locations have shown mixed results [9]. For the initialization and calibration of models, current hydraulic models require spatially distributed observation data at a scale proportionate to the model domain. Higher resolution data have been shown to lead to better simulation results [14]. These data may be available in various forms, including flood extents, water depths, or stream velocities. Data fusion techniques are required to ingest multiple sources and generate products for model use, especially as data are generated contemporaneously during an evolving flood event. This is particularly important in urban environments, where the progression of the flood may be propagating with complex geometries.

Several approaches were developed for leveraging remote sensing data for inundation modeling for model calibration [20], validation [21,22], and to improve model development [11]. Ref. [23] used MODIS and Advanced Spaceborne Thermal Emission and Reflection Radiometer (ASTER) images to acquire spatial extents of flooding to calibrate flood inundation areas using a distributed hydrologic model in Lake Victoria. Ref. [24] integrated river gauge data with physiographic data, such as DEM, land use land cover, and Landsat TM imagery, to accurately estimate flood damage in Pitt County, North Carolina. Ref. [24] demonstrated the effectiveness of the proposed methodology for flood extent mapping based on the reflection differences between wet and dry regions before and during the flood event. Additionally, they showed that this approach could help estimate water-inundated areas located under forested canopy and cloud cover. Aerial platforms, both manned and unmanned, are particularly well suited after major catastrophic events because they can fly below the clouds, and thus acquire data in a targeted and timely fashion. Ref. [25] used RGB photographs collected using a UAV, GPS river cross-sections, and DEM as data sources to generate a digital surface runoff model (DSRM). The DSRM was then used as input into HEC-RAS for hydraulic computations with steady and unsteady flows. Results showed that HEC-RAS performed best when the DSRM generated from the observed data was used to run the analysis and to validate model performance.

Novel information streams, such as social media, contributed videos, photographs, and text, as well as other open source data, are redefining situational awareness during

emergencies. When these contributed data contain spatial and temporal information, they can provide valuable volunteered geographical information (VGI), harnessing the power of "citizens as sensors" to provide a multitude of on-the-ground data, often in real time [26]. Refs. [27,28] demonstrated this through a series of papers and case studies in which crowdsourced photos and volunteered geographic data were fused together using geostatistical interpolation to create an estimation of flood damage in New York after Hurricane Sandy on October 2012. Refs. [20,29] used geolocated photographs attached to social media messages to estimate water depth using referenced objects in the pictures and generated flood maps. Water-level data were then used to validate the flood estimation model. Ref. [30], on the other hand, proposed a method to use Twitter data in conjunction with traditional data sources. They used the presence of tweets to start hydraulic model runs and assessed the model output by examining different water and velocity levels derived from social media.

Gradually comparing model output with various remote sensing platforms allows us to investigate different model initialization options and help pinpoint model setups and assess how modeling may be improved. This research focuses on utilizing multiple data sources generated during emergencies for the identification of flood extents. It proposes a methodology to develop a multiscale observation product (MOP) to generate a sequence of spatiotemporal maps by fusing multiple heterogeneous data sources. The methodology is based on a computational approach that allows for the mining of big data, information integration, and data fusion. Furthermore, due to the multitude of spatial and temporal resolutions associated with contributed data, this research assesses flood progression during the entire duration of the event. It incorporates data that are generated during the duration of the event and seeks to represent the contribution of each dataset. This, in return, has the potential to support traditional means of inundation mapping in near-real time and provide advantages for high-resolution flood hazard mapping. A comparison between the observational flood map derived from the proposed method with traditional numerical modeling approach demonstrates how the fusion of heterogeneous data sources could provide advantages for high-resolution flood hazard mapping.

The structure of the paper is as follows: Section 2 provides an overview of the hydrometeorological event and the study area. Section 3 provides a detailed summary of different data characteristics and acquisition dates. Section 4 discusses the methodology used to generate spatial and temporal flood extent maps from observed data. The results are presented and discussed in Section 5, and conclusions are presented in Section 6.

## 2. Case Study: 2015 Memorial Day Texas–Oklahoma Flood

May 2015 was one of the wettest months for the state of Texas, with an average precipitation of more than 23 cm (9 inches) falling throughout the state (NOAA 2016, NWS). For the first weeks of May, numerous counties statewide received above normal precipitation, resulting in saturated soils and raising the water table at least 5–10 cm above normal. During the last week of May 2015, on Memorial Day weekend, a slow-moving tropical storm system caused severe flooding across much of the southern counties of Texas and Oklahoma.

The sequence of storms triggered record-breaking precipitation anomalies and destroyed or severely damaged roads and infrastructure across multiple counties and major cities, including Dallas and Houston. The widespread, high-intensity rainfall across the states, along with the already saturated soils from previous rainfall events, led to a sharp increase in surface water runoff and overbank flooding throughout many regions. The duration and flood progression of this storm system was unusual, as most flood events in Texas usually last only a few days and are typically localized, affecting only one or two river basins [31,32].

The Dallas–Fort Worth area recorded more than 30 cm (15 inches) of rain. In the city of Dallas, nearly two dozen roads were closed, including major interstates, and homes were flooded as nearby lakes and the Trinity River system were overfilled with excess

water. Several levees were breached throughout the county, causing over USD 50 million in damage for the county of Dallas. Out of the numerous areas affected by the storm, a section of the Elm Fork Trinity River was selected as the area of interest (AOI). This was due to data availability and proximity to the state's capital, located 24 km (15 miles) northwest of Dallas, in Irving, TX. The entire study domain, represented in black in Figure 1, extends approximately 21 km and covers an area of 190 km$^2$. However, due to the differences in spatial resolution between different remote sensing platforms, a slightly smaller AOI was selected, highlighted by a red dashed outline in Figure 1. The AOI extends 15 km and covers an area of 150 km$^2$.

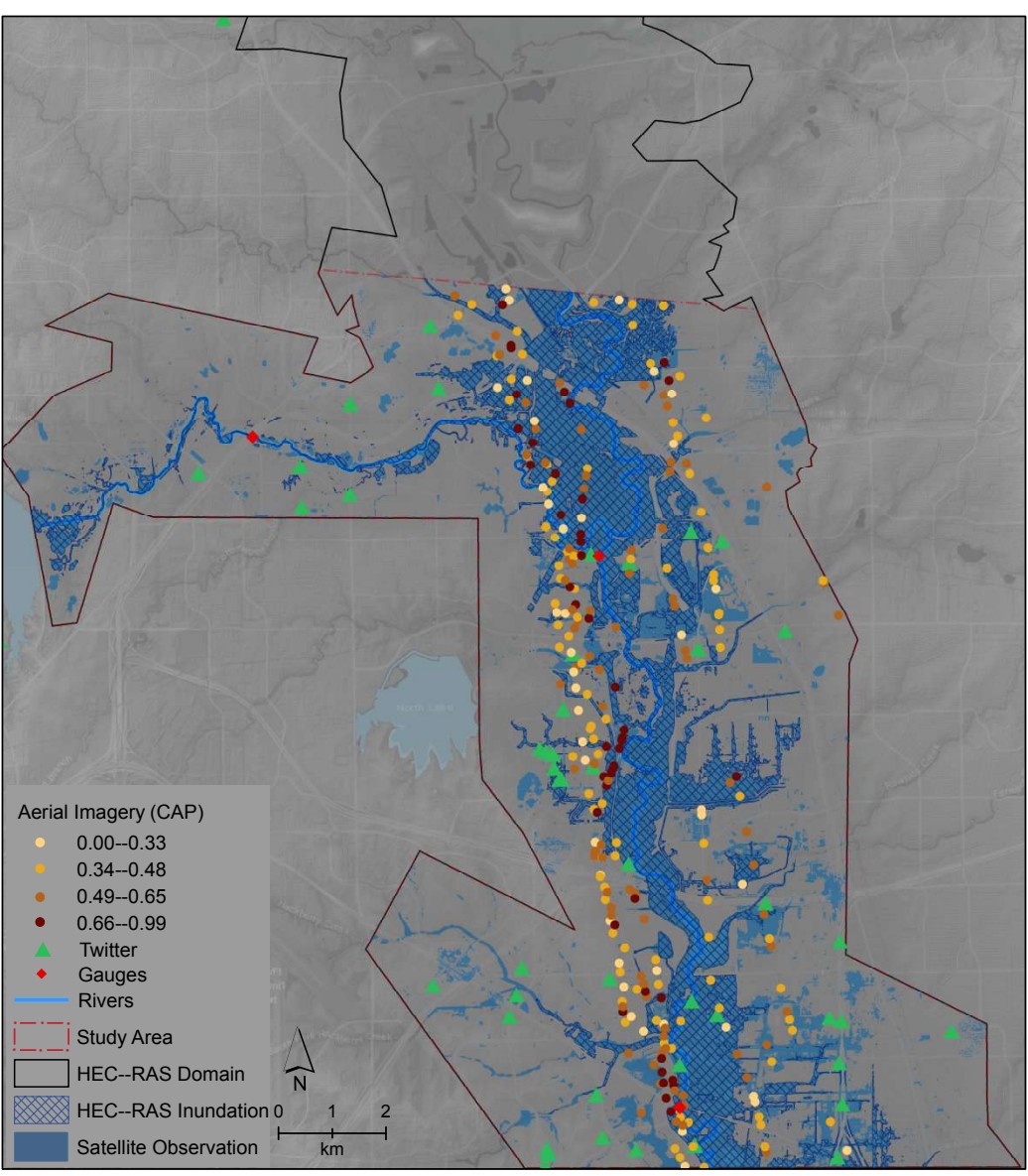

**Figure 1.** Study area showing the various sources of multispectral remote sensing contributed, and open source geospatial data collected and fused to estimate flood extent. The HEC-RAS model inundation is overlaid and represented with a hashed pattern.

The surrounding region outside the study area domain is a flood plain, mostly covered by light vegetation and slowly developing with upcoming urbanization. Monitoring this flood event from a hydrologic perspective was a difficult task due to the region's lack of stream gauges and generally poorly instrumented gauges (USGS, 2015). For this area, three river gauges along the Elm Fork River were available and were used to run the HEC-RAS

model. The gauges are shown with a red rhombus located east and south of the domain Figure 1.

## 3. Data

Multiple sources of multispectral remote sensing and open-source geospatial data were collected and analyzed to estimate the flood extent over the Trinity River area. All multispectral imagery was downloaded from the U.S. Geological Survey Hazards Data Distribution System (USGS HDDS). The HDDS is a unique web-based interface that enables users to search for satellite, aerial imagery, and other geospatial datasets available in near-real time based on a specific event. The web portal is designed to assist in disaster response by providing imagery and documents acquired before, during, and after an event. Datasets downloaded and analyzed for this event have multiple ranges of geometry, resolution, structure, accuracy, and date of acquisition. A summary of the data type, quantity, and acquisition day is shown in Table 1. A brief description of each data source and associated characteristics is provided below.

**Table 1.** Overview of all data sources summarized based on data type, quantity, and day of acquisition.

| Type | Source | Count | Acquisition Date |
|------|--------|-------|------------------|
| Satellite | | | |
| | WorldView-2 | 2 | 27 May |
| | WorldView-3 | 2 | 18 June |
| | SPOT-6 | 2 | 31 May |
| | Landsat 8 | 2 | 2 June, 18 June |
| Aerial Imagery | | | |
| | CAP | 273 | 21 May, 31 May, 1 June–5 June |
| Ground | | | |
| | Twitter | 97 | 21 May–29 May, 31 May, 1 June–4 June |

### 3.1. Remote-Sensing Imagery

Imagery from various high (smaller than 5 m) and medium (larger than 15 m) spatial resolution remote sensing platforms are frequently used for mapping areas of inundation. However, these data have limited applicability when mapping inundation over small-scale areas. Data with finer spatial resolution provide adequate information for deriving precise extent of flood inundation. For this study, both high- and medium-resolution datasets were used to help overcome the pixel distortion issues often present in medium-resolution platforms as discussed in Section 2.

Landsat 8 is the latest satellite by National Aeronautics and Space Administration (NASA), launched on February 2013. It provides multispectral data at 30 m resolution with a 16-day temporal resolution. Four Landsat scenes were selected to perform a spatiotemporal comparison over the duration of the flood event. Each image was chosen based on the percentage of cloud cover (<40) and extent of coverage for the study area. The following dates were used: 15 April, 1 May, 2 June, and 18 June 2015.

High-resolution multispectral imagery is also used in addition to the Landsat scenes. Commercial imagery from WorldView-2, WorldView-3, and Satellite Pour l'Observation de la Terre (SPOT-6) were chosen as they provide data more frequently, with a revisiting time of less than one day, and spatial resolution up to 31 cm with the panchromatic band. Two WorldView-2 scenes were used for 27 May 2015, two SPOT-6 scenes were used for 31 May 2015, and two WorldView-3 scenes were used for 18 June 2015.

Civil Air Patrol (CAP) imagery is an additional data source that has shown to be a valuable alternative data source when remote sensing data are not yet available [33]. The CAP is a congressionally funded, nonprofit corporation functioning as an auxiliary to the United States Air Force. The CAP conducts a variety of missions in support of federal, state, local, and nongovernmental entities, including search and rescue, disaster relief support, and aerial reconnaissance for homeland security. It collects hundreds of high-definition aerial photographs in the visible part of the EM and is used to assist in disaster response

for multiple emergency operations nationwide (i.e., search and rescue missions, forest fires, flood response documentation). A total of 278 images were collected for this study area.

### 3.2. Twitter Data

Twitter is one of the largest social networking sites, and it is widely used to share information through microblogging. These microblogs, or "tweets", at the time of the study were limited to 280 characters, so abbreviations and colloquial phrasing are common, making the automated content filtering challenging. Twitter is a very popular outlet during emergencies and disasters, and it is often used by government agencies and the public to disseminate information. The use of hashtags, words, or unspaced phrases prefixed with the sign # are central to the use of Twitter. They act as identifiers unique to Twitter and are frequently used to search and filter information. The creation and use of a hashtag can be established by any user who wants to create a concept category to share specific information about a subject [34]. Figure 2 depicts the various datasets used in this case study and illustrates the quantity and continuity of certain datasets in comparison to the traditional observation made at specific snapshots in time.

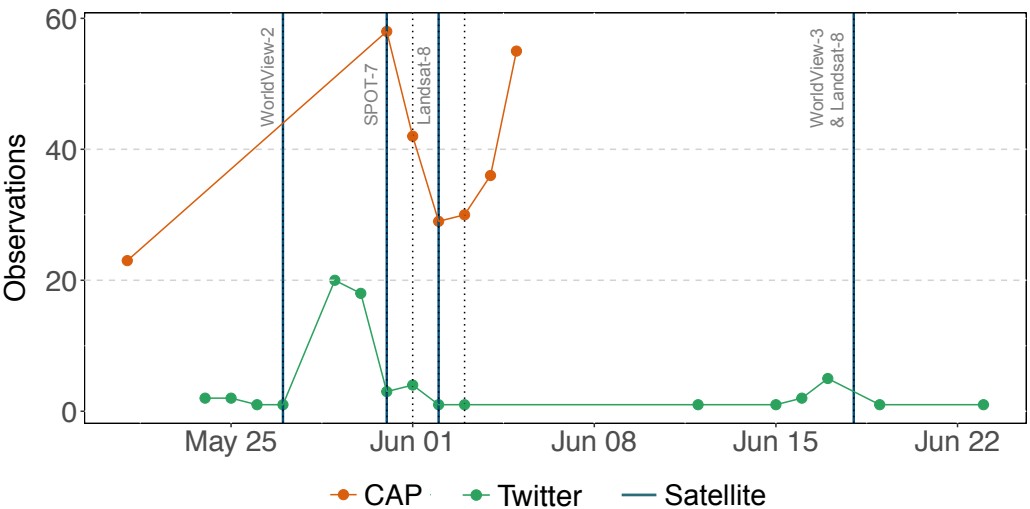

**Figure 2.** Summary graph of all data available during the 2015 Memorial Day flood. Vertical dotted lines represent dates for which more than one data source are available on that given day.

### 3.3. Hydrologic Engineering Center River Analysis System (HEC-RAS) Model Output

The hydraulic geometry of a river is an essential component for accurate model simulations and is generally dependent on the digital elevation model (DEM). A variety of numerical models have been developed as tools that aid in flood prediction, mitigation planning, and hazard assessment. The Hydrologic Engineering Center River Analysis System (HEC-RAS) is one of the most widely used modeling tools that is developed by the US Army Corps of Engineers. It has been present in the public realm for more than 15 years and has been peer-reviewed (HEC, 2010c). It is also widely used by many government agencies and private firms worldwide. For these reasons, HEC-RAS was selected for this study. A 5 m light detection and ranging (LiDAR) digital terrain model was used to accurately estimate the hydraulic geometry of the Trinity River. A series of hourly inundation extent outputs were generated using a 2D dynamic flood model for each day from 12 April 2015 to 14 June 2015.

## 4. Methodology

### 4.1. Overview

A methodology to develop a multiscale observation product (MOP) is proposed to generate a sequence of spatiotemporal maps by fusing multiple heterogeneous data sources. The goal is to generate 2D maps that could be used to assess the output of 2D numerical

models. Three main challenges associated with fusing numerous datasets are the varying spatial and temporal resolutions and the diverse data collection methods.

The 2D spatiotemporal maps were produced by analyzing each data source individually to estimate the water presence using the best available technique for that source. This step was repeated for days that the data were available. The water surface areas extracted from each source were resampled to a common grid and the maximum flood extent was computed using the sum of each day.

Lastly, the water surfaces in MOP were compared with the output of the numerical model using a fit measure that quantifies spatial differences. For this study, four different approaches were used to classify and identify water surfaces depending on the data source. These techniques are represented in Figure 3 and explained further in the following sections.

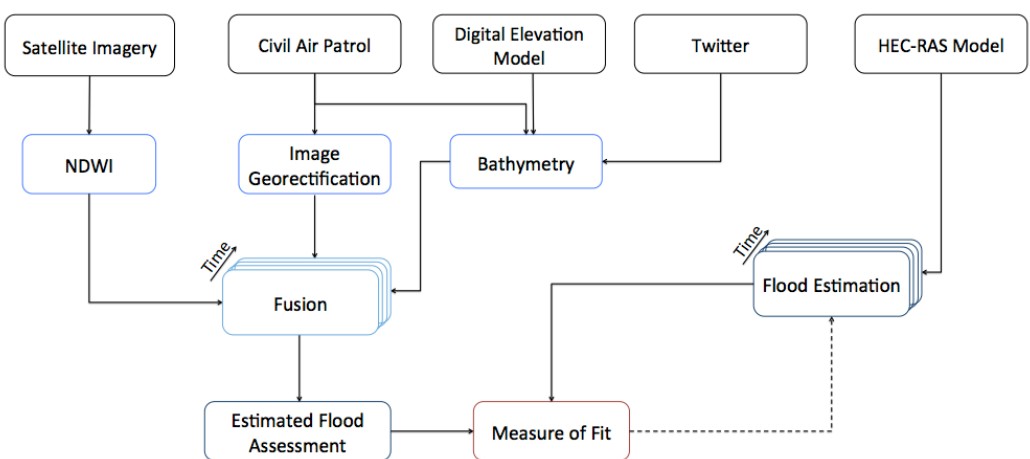

**Figure 3.** Diagram of the methodology used in this study to generate flood change detection maps.

### 4.2. Water Identification in Satellite Remote Sensing Imagery

Various image processing techniques have been introduced in recent decades for the extraction of water features from satellite data. The Normalized Difference Water Index (NDWI), first proposed by [35], was applied to delineate and distinguish the water extent over the Trinity River study area. The NDWI uses two bands, the NIR and green channel, of a multispectral remote sensing scene to classify open water features. The NDWI is calculated by the formula provided below, where ρGreen refers to the green band and ρNIR refers to the near-infrared band.

$$\mathrm{NDWI} = \frac{\rho\mathrm{Green} - \rho\mathrm{NIR}}{\rho\mathrm{Green} + \rho\mathrm{NIR}} \tag{1}$$

The NDWI is a dimensionless product, and values vary between −1 to +1. Values greater than 0.3 correspond to high water content and values less than 0.3 correspond to low water content and low fraction cover. This method was applied to all multispectral data available for the flood event. The threshold value was determined based on a study from [36] which found that a 0.3 value is best for separating water from a water and land mixture.

### 4.3. Water Identification in Aerial Imagery

Working with aerial imagery is a challenging task because these data are often acquired using conventional digital cameras, and only in the visible parts of the red–green–blue (RGB) spectrum. The aerial images were classified using an ensemble of a supervised decision tree machine learning classifier and a maximum likelihood classifier. Two classifiers were used because of the additional difficulty introduced by the lack of IR information. Furthermore, because of the lack of infrared (IR) data, a 2D wavelet transformation using the Haar mother wavelet was first run to extract texture information for all the images, and

thus expanded the search space beyond RGB. Then, for each transformed image representation (RGB + wavelet coefficient components), a decision tree learning algorithm was used to classify each pixel as water. A more detailed description of this method is presented in [33].

An additional challenge when working with aerial imagery is that these images often do not contain spatial reference information. Therefore, the images do not align properly with other geographical data available. In order to account for this limitation, each image was georectified (or georeferenced) to a geographic location or a map. This is commonly performed by taking an image in its original geometry and identifying a set of ground control points (known x–y coordinates). Ground control points are features such as road intersections, bridges, buildings, trees, etc., that are present in the aerial imagery and also have known coordinates in the real world. Once the points are identified, a polynomial transformation is applied to distort and match the original (unprojected) raster with the projection of the map. In this study, the ESRI ArcGIS projective transformation (http://desktop.arcgis.com/en/arcmap/10.3/manage-data/editing-existing-features/about-spatial-adjustment-transformations.htm accessed on 1 September 2015) approach was chosen to transform oblique imagery directly from aerial photography. After the photographs were individually assigned a coordinate system, they were mosaicked into one new raster and the water extent was identified for each date.

### 4.4. Water Identification in Twitter

This step consisted of generating a flood inundation extent by integrating ground data with topography, which is one of the most important characteristics in flood scenarios. The nature of these data are different from data used to generate the previous flood extent maps, as they comprise sparse points for each day. The points were used to help identify the presence or absence of flooding in a localized region around the coordinates of the tweet. Twitter posts were filtered based on the geographic location that the messages were shared from and their associated hashtags relevant to each event. Specifically, the x–y coordinates associated with each tweet were used to generate a 500 m buffer over the DEM centered at the x, y location. Within the buffer surrounding the Twitter point, the lowest height value was chosen and used as a threshold. Areas that were lower than the threshold value were flooded, while areas above the lowest elevation remained nonflooded. Each localized flood area was merged with the neighboring areas to generate a complete flood inundation for the entire domain. This process was repeated for each data point independently for all days that Twitter data were available. Additionally, the same process was duplicated for the CAP dataset, to generate the CAP.DEM product.

### 4.5. River Analysis System Preprocessing

Numerical models, such as HEC-RAS, have the capability of producing continuous outputs at different time steps throughout the simulation period. In this study, hourly HEC-RAS outputs were generated for each day spanning a two-month period from 18 April 2015 to 14 June 2015. The hourly outputs were aggregated to a single maximum flood extent for each day over the two-month period. These outputs were used for comparison purposes with the days for which observational data were available.

### 4.6. Data Fusion

Data fusion is a process by which data from multiple sources with different spatial and temporal resolutions are integrated together to improve information for decision making [37]. This approach is used throughout many disciplines. Although there is a wide selection of techniques and processing possibilities that have the ability to combine information, all methods incorporate an ordered hierarchical approach that takes data collected from multiple sources and represents them as an assimilated interpretation of a study area [37].

Flood extent layers were created from available remote sensing data, digital elevation models, and ground information to generate temporal flood assessment maps. Since data vary in spatial resolution, all data were resampled using the nearest neighbor algorithm to a uniform 5 m × 5 m grid to match the DEM resolution used in the HEC-RAS model. In the nearest neighbor algorithm, the center value of a odd-sized mask (e.g., 3 × 3, 5 × 5) is assigned the value of the average of the values in the mask. A three-day window centered at a specific date is used to increase the temporal resolution of the MOP results due to the sparseness of the observations available. This is shown in Figure 4, where the purple points represent the original acquisition day for the data and the gray shaded areas represent the data extended within the temporal window. This approach extended the data available for analysis from only six days, as originally shown in Figure 2, using vertical lines, to a total of 20 days. By combining datasets of different spatial and temporal resolutions, it allows to generate inundation maps that have a higher temporal resolution than a single snapshot in time. The maximum extent for each time step was calculated for comparison with the HEC-RAS model output.

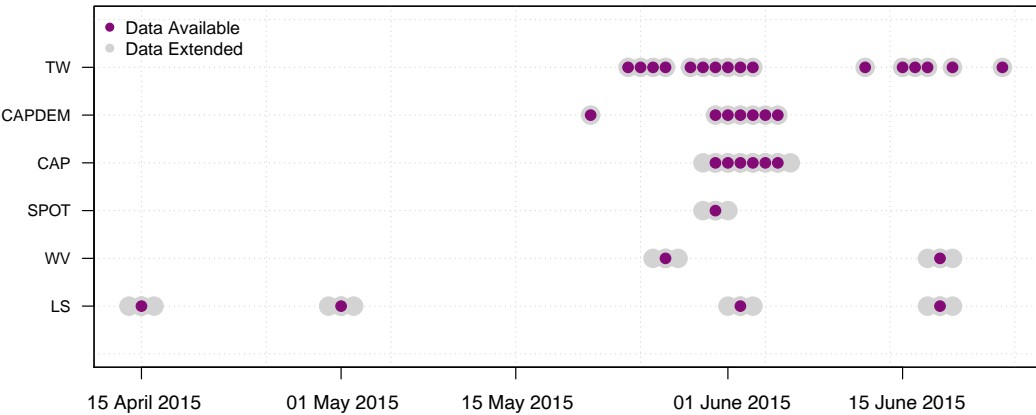

**Figure 4.** Diagram showing the temporal data extension after the data fusion method was applied. Purple dots represent the original day of data acquisition. The light gray dots represent the days added after data are extended via interpolation.

### 4.7. Measure of Fit

The comparison of mapped water features from the remotely sensed product to areas the model predicts as flooded is a common practice used to quantify the deviation of simulated results from observational data and thematic map accuracy. This is carried out using three metrics: measure of fit, omission error, and commission error.

The measure of fit ($F^2$) statistic represents the inundation extent of the observed and modeled data, respectively. This was calculated using the number of flooded pixels in both the observed remotely sensed products and the predicted HEC-RAS model output, over the total number of flooded pixels, calculated by taking the union of the flooded pixels in both the observations and model ($1_{obs} - 1_{mod}$).

The following equation was used to estimate such measurement, where $A^{obs}$ is the total number of water pixels from MOP, and $A^{mod}$ refers to the total water pixels predicted by HEC-RAS.

$$(F^2) = \frac{A^{obs} \cap A^{mod}}{A^{obs} \cup A^{mod}} \qquad (2)$$

The areas of omission and commission are two additional metrics used for comparing estimated water features from each output. These were used to measure the underestimation (omission (O)) and overestimation (commission (C)) of the simulated flood pixels (HEC-RAS) in relation to the observed water pixels from MOP. Therefore, omissions represent the pixels observed as wet but simulated as dry ($1_{obs} - 0_{mod}$). Commissions represent pixels simulated as wet but observed as dry ($0_{obs} - 1_{mod}$).

Omission error data were based on the requirement that the map labels matched the reference labels, whereas the commission error data were based on the requirement that the reference labels matched the map labels.

$$O = \left( \frac{A^{obs} - P_e}{P_u} \right) \times 100 \tag{3}$$

$$C = \left( \frac{A^{mod} - P_e}{P_u} \right) \times 100 \tag{4}$$

where $P_e$ is the number of common flooded pixels between the observed and predicted, and $P_u$ is the total number of pixels observed and modeled as water from MOP and HEC-RAS.

## 5. Results

HEC-RAS is the underlying model used for most flood simulation case studies conducted by FEMA to generate flood maps. Flood modeling analyses are conducted with the objective of determining the maximum possible flood extent. For this case study, the maximum inundation from all observation data and the calibrated HEC-RAS model output is shown in Figure 5.

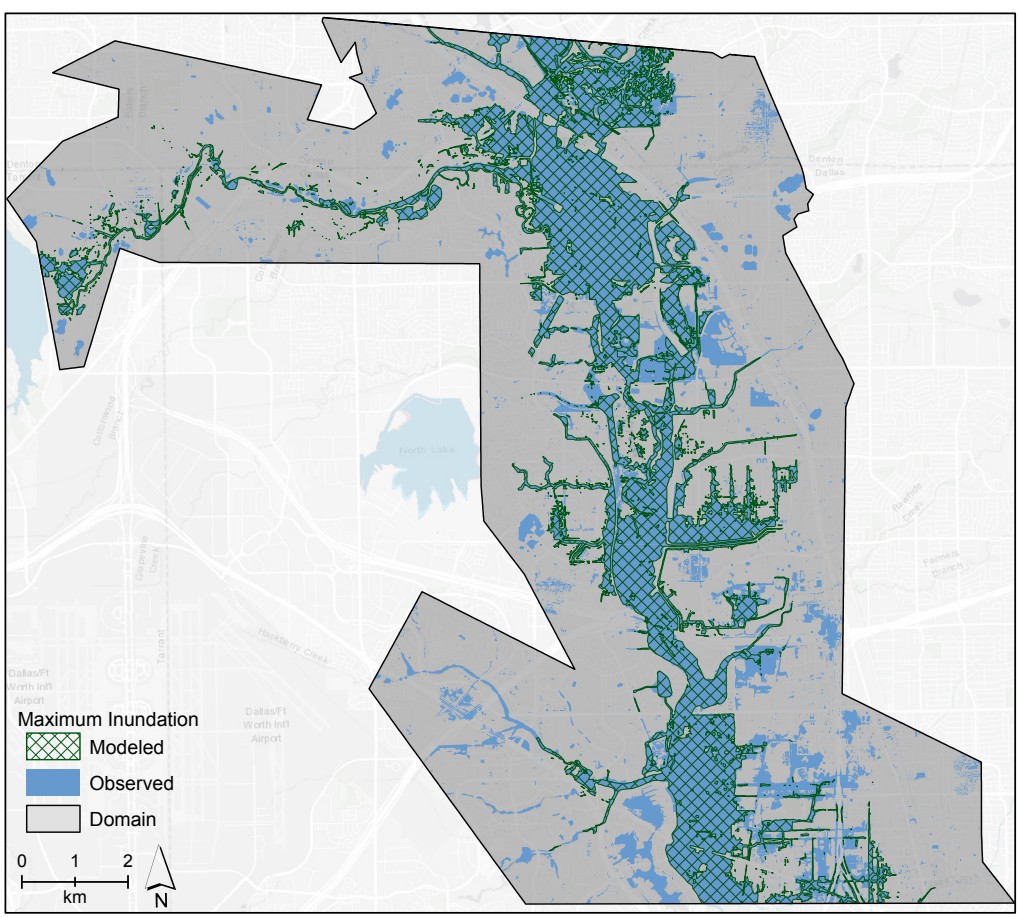

**Figure 5.** Water surfaces for each day are aggregated together to represent the maximum flood extent possible for the entire event. Inundation of the MOP is shown in light blue and the simulated HEC-RAS model output is overlaid in hashed green.

The maximum extent was determined by taking a single day of observation data for the 20 days that the original data were extended to and consolidating it together to estimate the extreme possible flood inundation (worst-case scenario) for the duration of the event based only on the observed data. The observed water surfaces generated from MOP are

represented in light blue. This step was repeated for the simulated HEC-RAS output for the same 20 days and is represented by the hashed green pattern. It is important to note that floods are dynamic in nature, and throughout the progression of an event, water depth and velocity are prone to change. Using either the simulated or observed maximum flood extent may not be a true representation of the actual extent at a specific time [12]. This is likely to occur in urban environments where different city areas may experience the effects of a flood event at different times.

This analysis compared individual days that observation data were available with water surfaces simulated from HEC-RAS. In comparison with observation data, numerical models are capable of providing large amounts of data as output, often continuous in nature over the domain area. Fusing heterogeneous data with different spatial and temporal resolutions expands data accessibility and allows for continues flood inundations monitoring at finer resolutions than various snapshots in time. This allows to assess both outputs spatially and temporally before, during, and after a flood event, as shown in Figure 6. The figure shows the total inundation area in km$^2$ over the duration of the flood event derived both from the observed data (blue line) and modeled by HEC-RAS (gray line). Different observation data available are represented with a series of symbols as well as which data source is used to estimate the inundation area each day. A gray dashed line is used as a baseline to represent the total area of water surfaces present during normal conditions. Note that CAP and TW are often below the baseline because they only observe a portion of the domain.

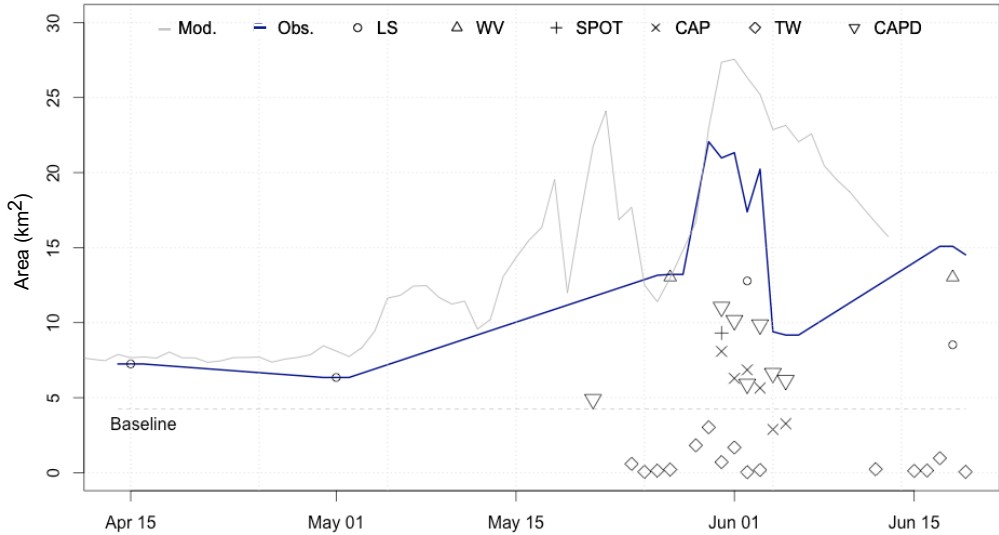

**Figure 6.** Data trends of the total area estimated from the remotely sensed product and simulated inundation over an eight-week time period. The total area of observed water is shown with a blue line and each dataset is represented by a series of symbols. The simulated water is represented by a gray line and the normal water area is shown with a dashed line.

The line trends of the total area observed from the MOP and predicted demonstrate an overall comparable pattern. The first day of the study period, both observed and predicted, shows an area of approximately 7 km$^2$. As the event progresses, both outputs show an increase of water presence inside the domain. This is particularly evident during the peak of the event, 31 May to 1 June, where both observed and predicted line trends peak at the same time for the same days. As the flood begins to recede, the total area of water presence begins to decrease.

Additionally, Figure 6 shows that the total areas simulated by the HEC-RAS model are almost always greater than the MOP areas from the remotely sensed data. This is due to two main reasons: one is that engineers open and close gates located along reservoirs or lakes in order to maintain a specific water level within the water bodies. The release of

water is carried out gradually over the terrain based on how much water flow the area can sustain. However, data that are reported and used as input for the HEC-RAS model do not capture this slow release; instead, sharp increases are caused and can be seen throughout the domain over a two-day period. This effect is shown in Figure 7, in which the image on the left represents the water surfaces simulated by HEC-RAS for 16 May 2015, and the image on the right shows the flood simulation for 18 May 2015. It can be observed that over this two-day period, the water is slowly propagating from the edge of the domain towards the center.

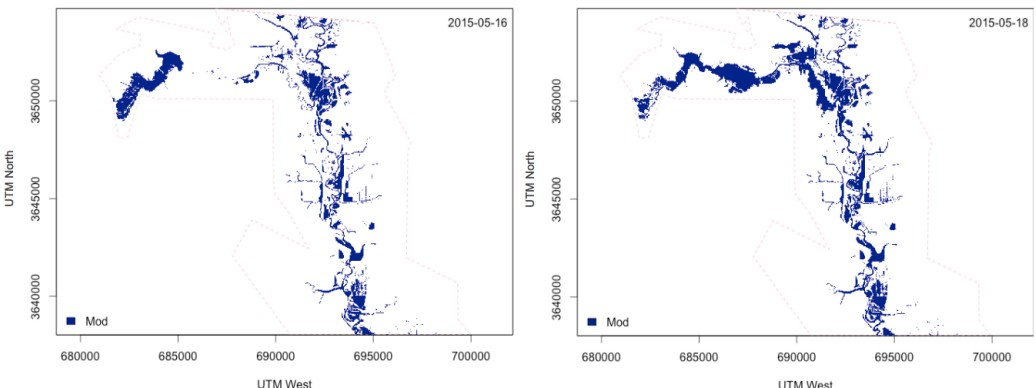

**Figure 7.** Water surfaces simulated from HEC-RAS for 16 May and 18 May are shown. Water can be seen propagating towards the center of the domain over a two-day period as water is being released from the nearby lake.

The second reason why HEC-RAS has higher areas estimated than the MOP is due to the underlying characteristics of the observational data used each day. The only data available near 15 and 18 May are CAP. These data are spatially distributed along the Elm Fork Trinity River and do not cover the creek left of the Trinity River. This can be seen in Figure 1, in which the CAP is represented by the shaded color of yellow and dark orange. This is also true for the sharp decrease in area shown by the observed water features for 3 June, where the only data available are the CAP aerial imagery and Twitter. Although these data have the advantage of having a high temporal resolution, they are often spatially sparse, or, in the case of CAP, are assigned to collect data only over a specific geographical area, while the opposite is true for satellite remote sensing imagery, which are characterized by high spatial but low temporal resolution. Although the CAP data may be spatially constrained over specific areas, their temporal resolution helps to fill in the gaps when satellite resolution from platforms is not sufficient. This is shown and further explained in the following section and depicted using Figure 8.

Another reason why HEC-RAS may have simulated a larger flood extent compared to the observed dataset is because of errors in the main channel bathymetry, as represented by the DEMs. While DEMs (such as National Elevation Dataset) are widely used for flood modeling, these datasets do not accurately represent river channel and large water bodies, but represent them as flat surfaces, resulting in underestimating the flow conveyance potential [38,39].

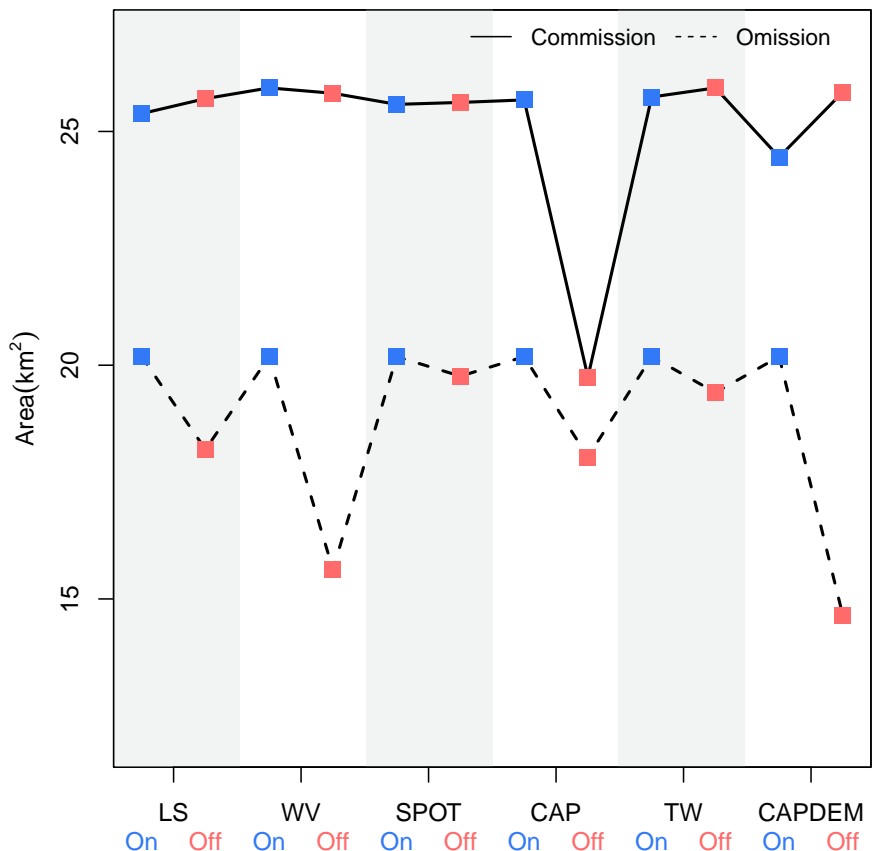

**Figure 8.** Depicts the contribution of each data source to the commission and omission errors. Data included in the error are represented by blue squares, while data omitted are shown using red squares.

### *5.1. Error of Commission and Omission*

Estimating possible flood inundations using individual datasets that vary spatially and temporally allows us to visualize and understand how much each observational dataset contributes to the overall water estimation individually. This is performed by computing the omission and commission errors that occur when each dataset is either present or absent from the MOP. Before calculating the omission and commission errors, the sum for a single data source being present is calculated for every possible data combination. This step is repeated, but instead of including all data, the same source is omitted, then the sum is calculated for each data combination possible without that one data source. Once the sum is generated for each data, the union and intersection are obtained in order to calculate the overall omission error and commission for each data source. Since for this case study, we have a total of six different observation data sources, each one is turned on and off; thus when the sum is estimated there are a total of 12 layers that are used to estimate the commission error and 12 layers used to calculate the omission error. The total 24 errors are shown in Figure 8. Each error value shows the overall area (km$^2$) that each data source overestimates or underestimates during the duration of the flood event; this is used to represent the overall, on average, trend of error.

Commission error is represented by a solid line while omission error is represented by a dashed line. Each individual data source is listed along the *x*-axis together with when it is used (on and off) to estimate the errors. Data being encompassed in the error are represented by blue square symbols, while data omitted are shown using red square symbols. When looking at Figure 8, it is important to note the dimensions that each data layer represents. In this case, there are three parameters: 1. The total area of observation that each dataset identifies as possible water surfaces; 2. The water surfaces that are being observed as well as predicted (the intersection of MOP and HEC-RAS); and 3. The number of days that are removed as each data layer is omitted.

The commission error trend line in Figure 8 depicts that overall all the satellite remote sensing data (LS, WV, and SPOT), whether they are being used or not, the commission error with respect to the predicted water features by HEC-RAS slightly increases or stays the same. This is because the areas of intersection (water features seen by both MOP and HEC-RAS) change as each data layer is added or removed. This effect is most noticeable when the CAP imagery is removed or turned off from the MOP, causing the commission error to decrease. This shows that normally there is a large overlap between the water features predicted by HEC-RAS that are also observed from the CAP images compared to any other dataset. Additionally, this implies that although the CAP images are collected along a specific portion of the domain and may be sparse, in this case, they are a very important data source when fused with other observational data.

The omission error, represented by a dashed line, is used to show, on average, the relative contribution of each dataset individually when compared to MOP. It can be noted that every time a data layer is added or removed, the omission error increases or decreases. This is expected because as each data layer is being removed, a number of observed water surfaces unique to that data source are also being removed. Additionally, along with removing the data layer, all the days that the data source is available are also being removed from MOP. This is also true when all data layers are turned on, as the omission error is relatively the same.

The temporal component based on the number of days that are added or removed with the data layer as well as the water surfaces that are unique to a single data source can be best seen when comparing the WV and CAP.DEM layers with the SPOT and TW sources. For this, the WV and CAP.DEM layers contribute the most omission error in comparison to the other datasets as they contain water surfaces that are not observed by any other data layer. This means that as WV and CAP.DEM are removed, there are less water features that are predicted by the model. In contrast to the WV and CAP.DEM sources, on average, SPOT and TW provide very little information to the MOP. This is because for the days that SPOT and TW data are available, there are several other data sources which are fused for those days; thus, the other sources help augment the absence of SPOT and TW but not for the WV and CAP.DEM data.

Due to the fact that flood events and the data available to observe them change significantly over the duration of an event, it is important to understand the contribution of each dataset. On average, when a specific data layer is removed, the days that the data covers also have to be removed. However, throughout different flood phases, the uniqueness of the data source may increase severely as seen by the CAP, WV, and CAP.DEM. Yet, in other circumstances where other available data with similar characteristics are available, then some datasets may not be as unique, as shown by WV and TW. Therefore, the uniqueness of a data layer is a very big factor in whether or not the errors of omission and commission areas increase or decrease.

*5.2. Visualizing Areas of Commission and Omission*

To represent calculated areas of commission and omission estimated by MOP and HEC-RAS, the areas observed and omitted from each data layer are individually plotted. This is performed to show the areas of data agreement and to spatially represent the contribution of every dataset on average as each layer is turned on and off. In addition, the difference between the data sources is presented to help visualize the various water regions of omission and commission areas. Figure 9 represents the areas of omission, in coral, and commission, in purple, for all data sources used in MOP over the domain. Additionally, using a blue hashed pattern, are highlighted the regions of water surface intersection between the MOP and HEC-RAS output. The differences of relative areas of omission and commission that occur when these data are employed or disregarded are shown in the third column.

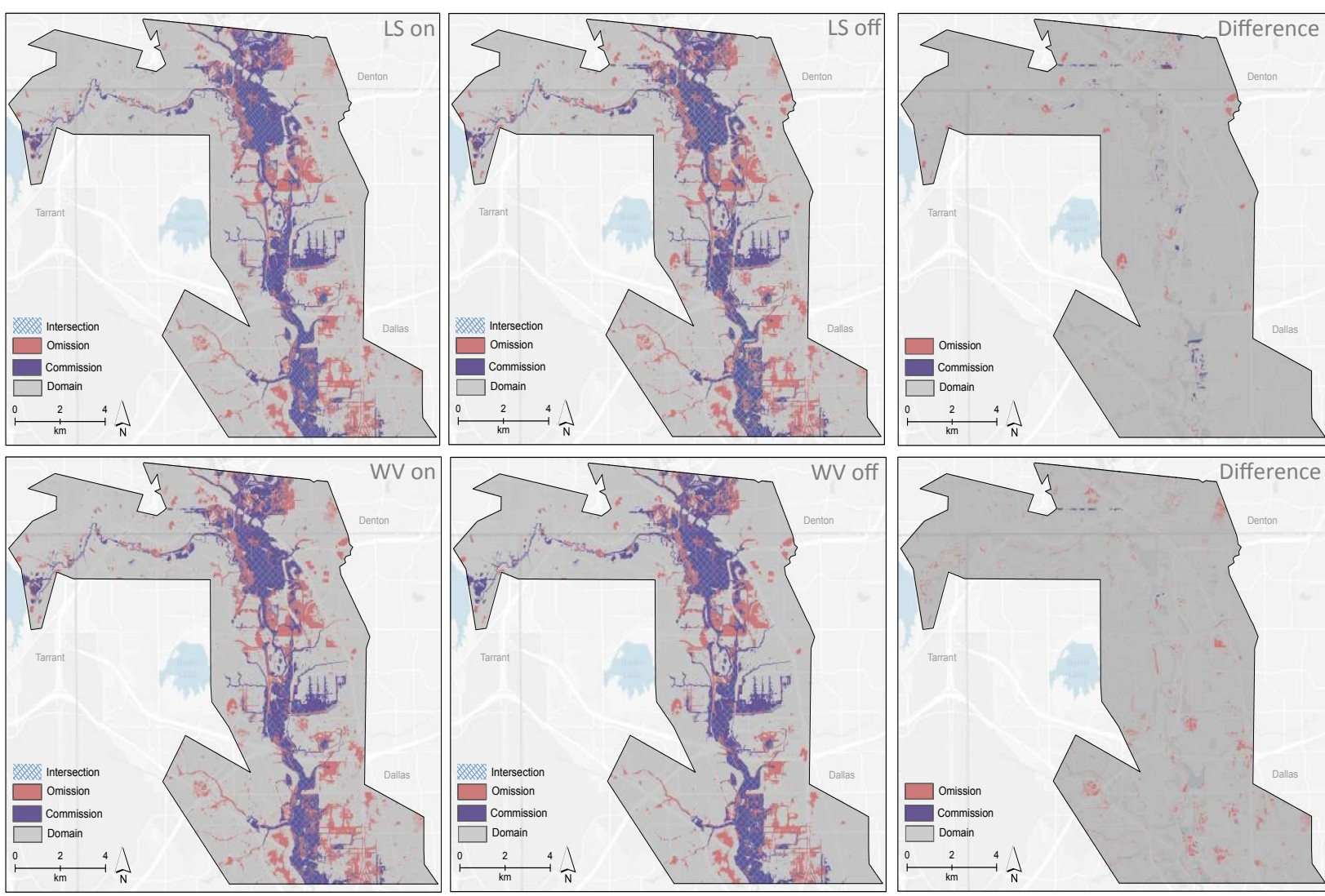

**Figure 9.** *Cont.*

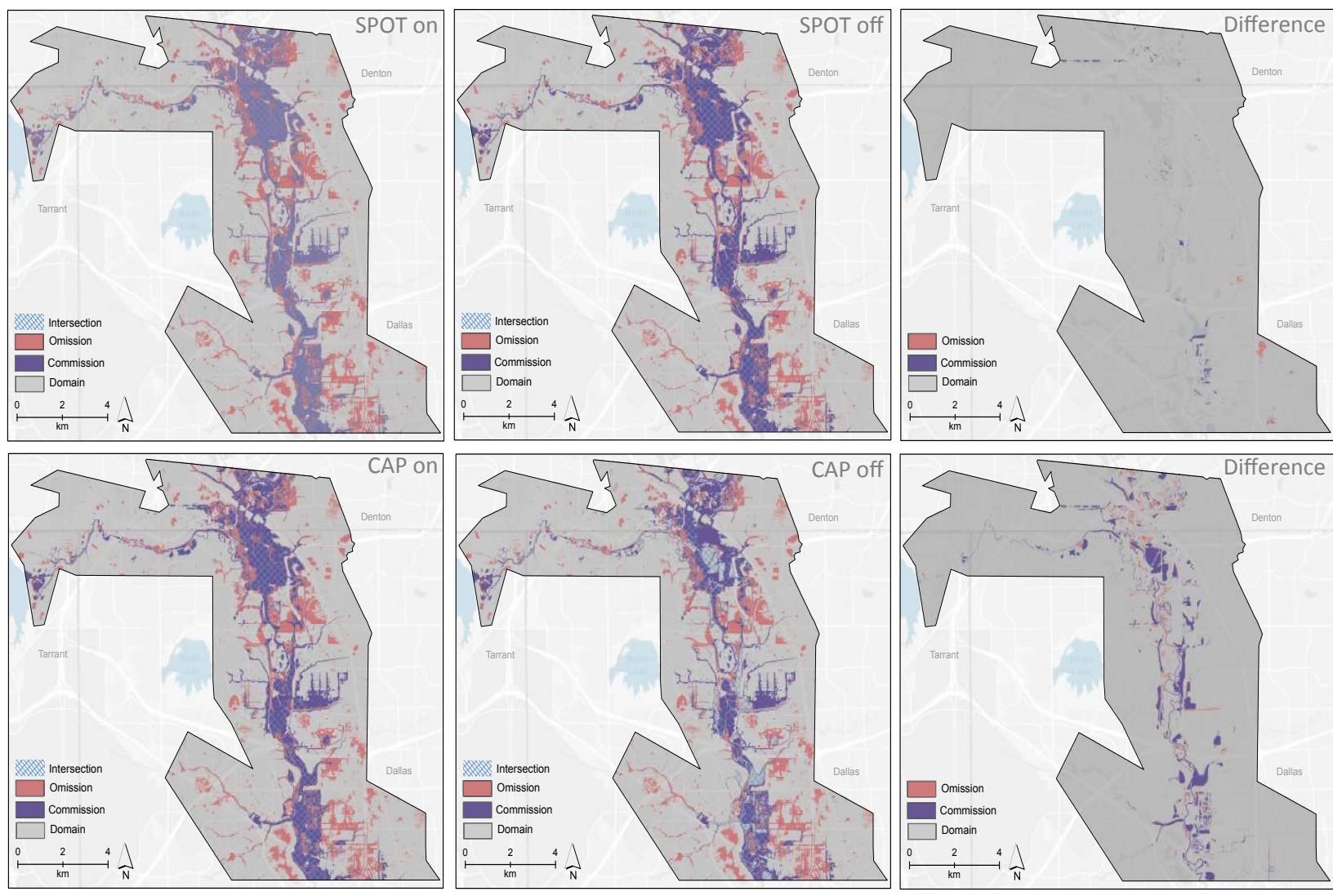

**Figure 9.** *Cont.*

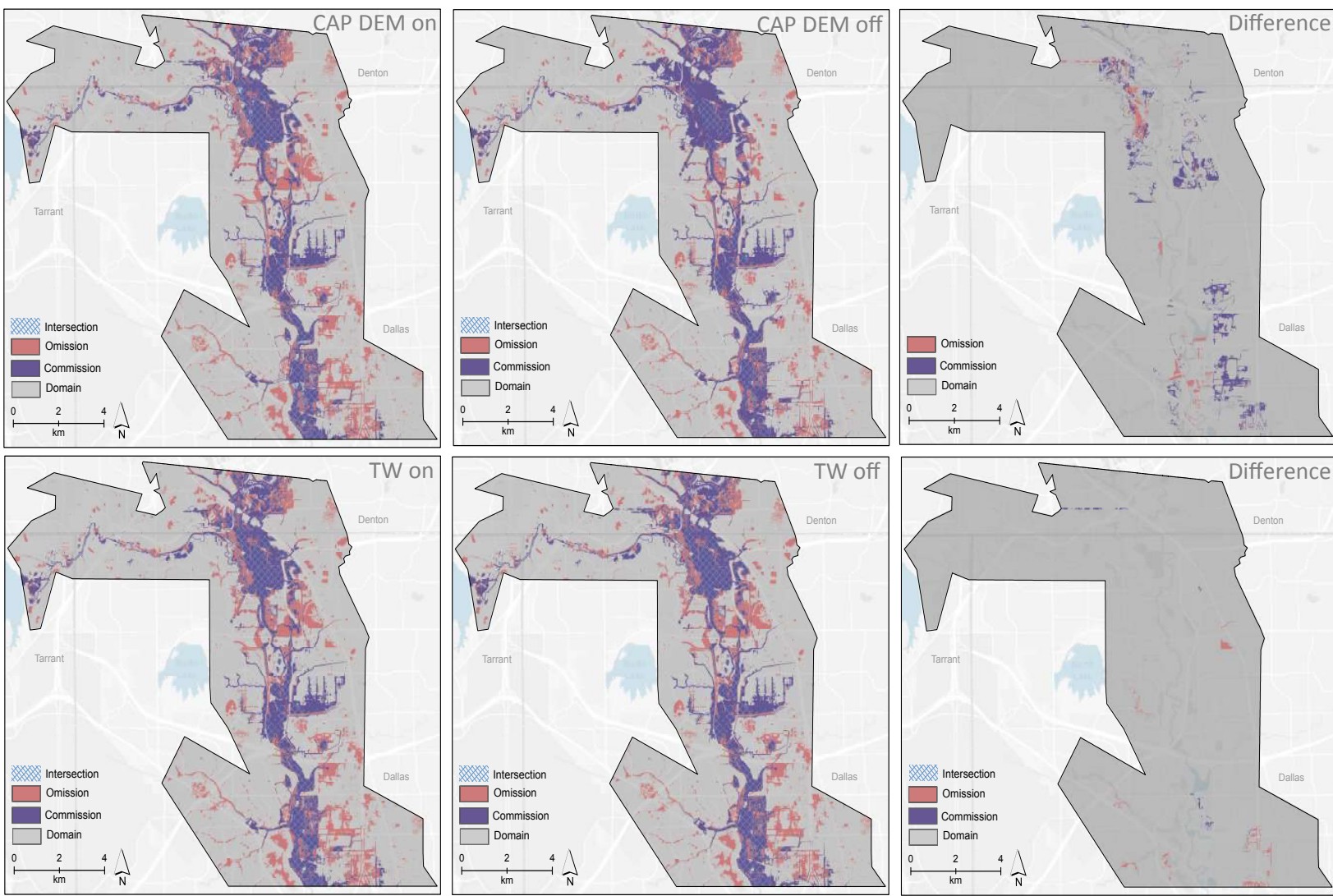

**Figure 9.** Areas of commission and omission are shown in comparison to the HEC-RAS model output. LS and SPOT observations are displayed on the top row of the figure, and WV, CAP, and TW observations are below. The intersection between the observation data and HEC-RAS is shown with a light blue hashed pattern.

Looking at the differences between each data layer, the patterns of average areas highlighted in Figure 8 and discussed in the previous section can certainly be noticed. For example, the areas of commission shown by the satellite observations (LS, WV, SPOT) are all relatively the same. Yet, the water surfaces observed by CAP and CAP.DEM throughout the domain are most dominant with commission errors. The areas of omission which were shown to occur when WV and CAP.DEM data were turned off are also displayed. In fact, it can be observed that the areas in which the omission occurs indeed have the most omission compared to the other data available. The different areas of intersection between every observation dataset with the HEC-RAS output are approximately similar, with the exception of the CAP dataset. This implies that during the dates that CAP imagery are available, they are able to observe water bodies that HEC-RAS is not able to predict as flooded, resulting in the largest area of commission than any other dataset used for this study.

### 5.3. Daily Flood Estimation

During emergencies, it is necessary to analyze large amounts of heterogeneous geospatial data in a collaborative environment and in a timely fashion. It is critical for decision-makers and emergency responders to have access to timely actionable knowledge regarding preparedness, emergency response, and recovery before, during, and after a disaster [40]. The use of maps during an event has been shown to be vital, as they are the quickest method for locating assets in a specific geographic area without having to separately reference large amounts of data. Additionally, estimating possible flood inundation for individual days for the duration of a flood allows to visualize changes that may occur throughout the progression of the flood event.

In this regard, cumulative and temporal flood extent maps were prepared. These are shown in Figures 10 and 11, which represent, respectively, the flood inundation during the beginning of the flood event and at the peak. Both figures are divided into four quadrants displaying different aspects of the flood event. The top left quadrant represents the flood trend areas estimated from MOP and HEC-RAS for a single day. The top right quadrant shows the areas of omission, commission, and data intersection throughout the domain, while the lower two quadrants display HEC-RAS inundation and MOP spatial extents. Figure 10 represents the flood inundation for 15 April 2015, which is one of the first days of the simulation.

Estimating flood inundation on a daily basis using heterogeneous data enables a better understanding of how the MOP compares with the simulated water surfaces from the HEC-RAS output each individual day. The goodness of fit measurements for each day of the analysis is summarized by Table 2. On 15 April 2015, a 43.58% agreement between HEC-RAS and MOP is calculated; this can be noted by referring to the total area estimated in the first quadrant or by looking at the data intersection in the second quadrant of Figure 10. A 27.59% omission error by MOP and 28.83% commission error by HEC-RAS is shown in the second quadrant and are highlighted by coral and purple colors. Figure 11 shows flood inundation extent at the peak of the event on 1 June 2015. The goodness of fit between HEC-RAS and MOP is 51.28%, and the omission error by MOP is 19.47%, while the commission error of HEC-RAS is 29.24%. An average of 42% fit is calculated for the entire duration of the event between MOP and HEC-RAS.

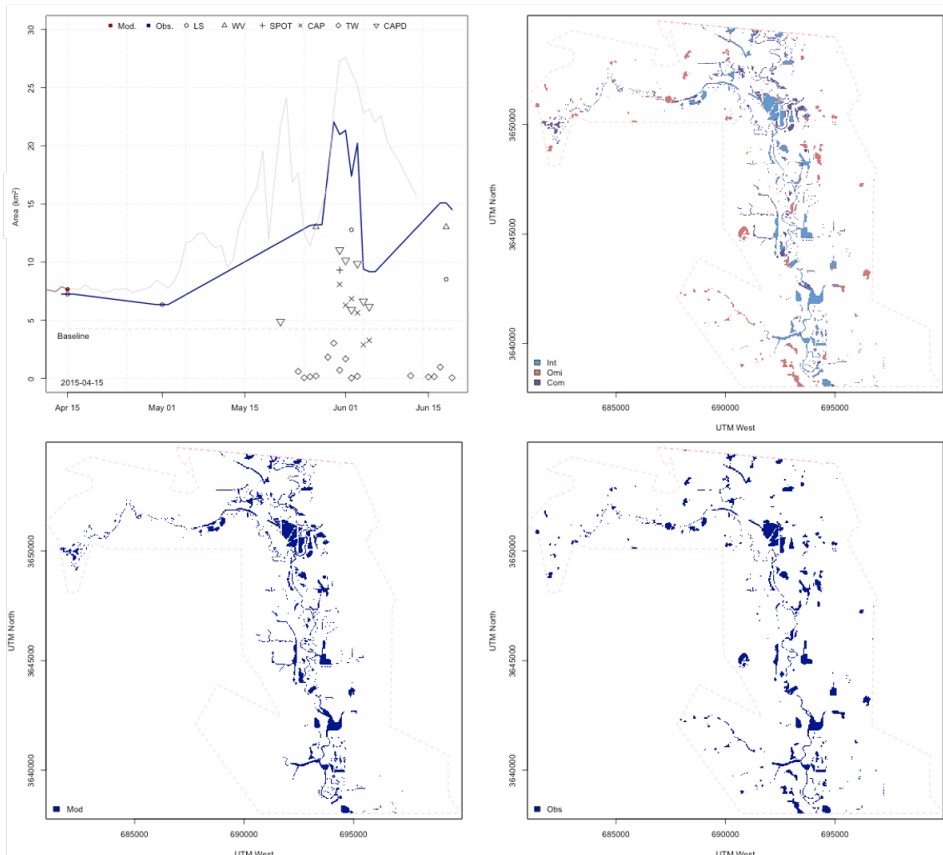

**Figure 10.** Flood inundation for 15 April 2015 estimated by MOP and HEC-RAS.

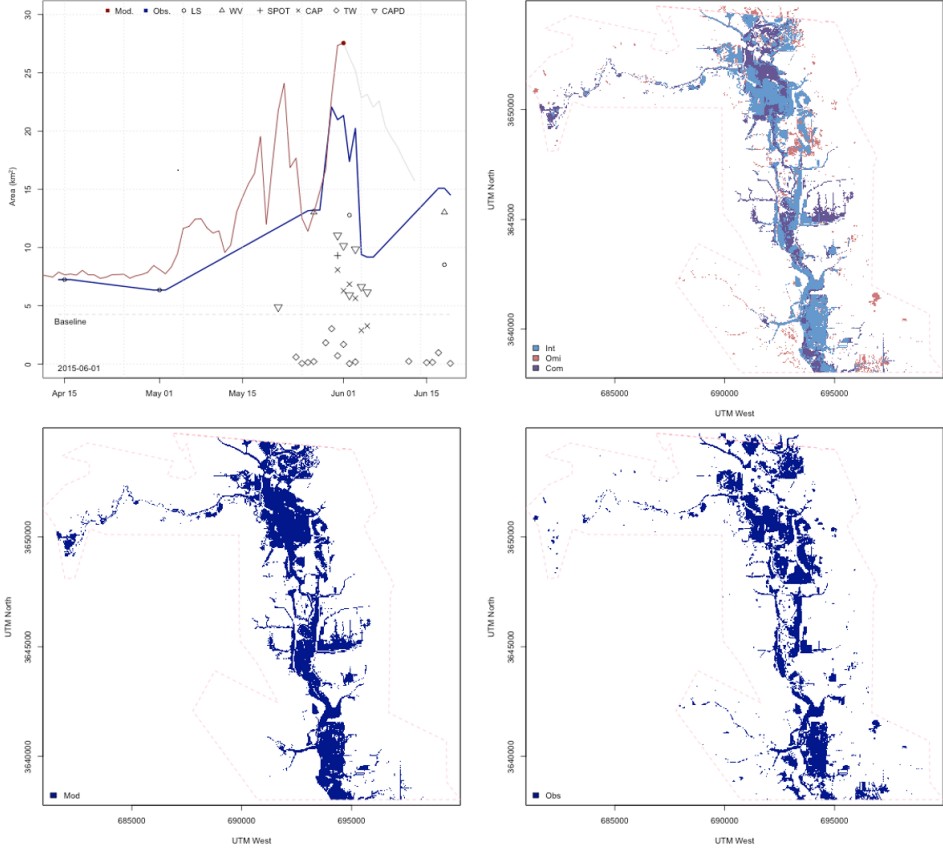

**Figure 11.** Flood inundation for 1 June 2015 estimated by MOP and HEC-RAS.

All phases of emergency management depend on data from a variety of sources. The appropriate data have to be gathered, organized and displayed logically to determine the size and scope of emergency management programs in order to respond and take appropriate action. A pressing question when working with a series of heterogeneous datasets is which data are needed the most at any given time. This is extremely important in times of emergency, as officials must act quickly and cannot afford lengthy analysis. For this case, the total area observed is used to estimate which dataset is most useful for any given day. This is determined by calculating the total area of omission or commission based on the data available for each day and all the data combinations possible. A threshold value, based on the top 30% of calculated areas, is set to represent which data are needed in order to obtain an approximate area of omission for our study region. Figure 12 is used to suggest which data should be used and which should be omitted based on the threshold value of an area less than 5 km$^2$ (this is approximately 30% of the maximum value of seven).

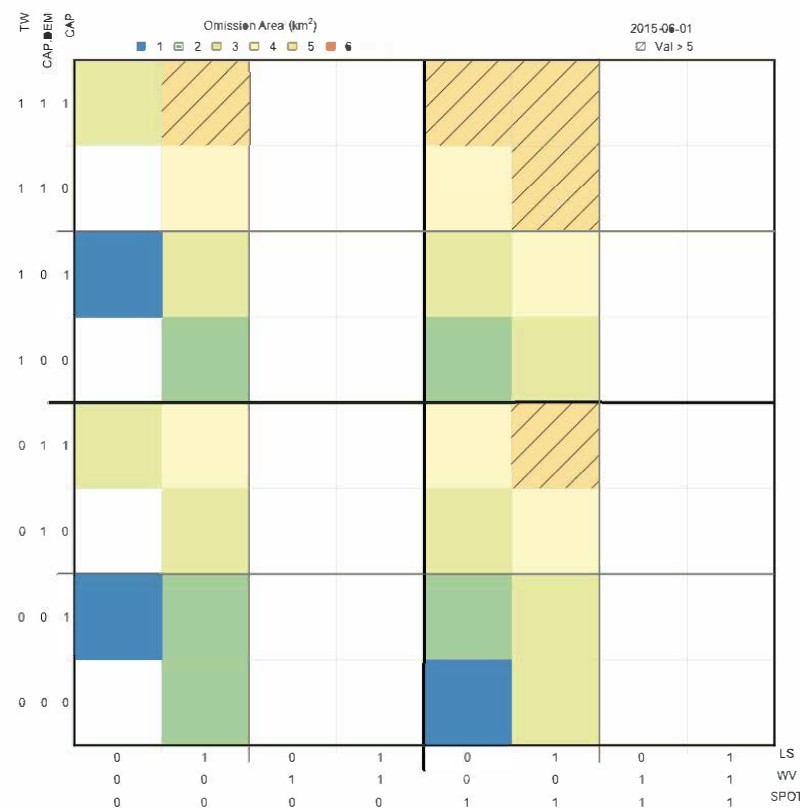

**Figure 12.** Areas of omissions for all data combinations available for 1 June 2015.

Figure 12 represents the areas of omissions based on the data combinations from all data available for 1 June 2015, a day of the flood peak. Satellite observations are listed along the x-axis, and social media with aerial photos are along the y-axis. Dataset combinations that estimate an omission area larger than the threshold value are emphasized by a cross-line pattern. For this example, it can be inferred that both satellite data are not necessarily needed as long as any of the other datasets are available. Either SPOT or LS are suitable as long as they are fused with other data. Additionally, looking at SPOT and LS individually, it can be determined that SPOT has a smaller area of omission than that of LS for this single day. This concept can also be used to highlight which data are needed to estimate areas of omission greater than 5 km$^2$. In this case, the options are a lot less limited and cannot rely only on a single dataset, but there must be access to specific data combinations such as the ones shown with a cross-lined pattern in Figure 12.

Similarly, Figure 13 shows which of the data combinations are used to represent the total areas of commission based on the data combinations for 1 June 2015. The figure

shows that for any combination in which LS or SPOT are present in the fusion, the area of commission is low ($\approx$8–10 km$^2$); however, when they are individually used, the areas of commission are higher. This is specifically true for SPOT. This implies that SPOT alone has a larger area of commission than LS and is a data source that could be used independently to observe an area greater than 15 km$^2$. Other data combinations that only have an area of error for commission greater than 15 km$^2$ are the CAP and CAP with Twitter. This implies that these datasets are the only combinations that provide a commission error greater than 15 km $^2$ for 1 June 2015, and the CAP is an important data source.

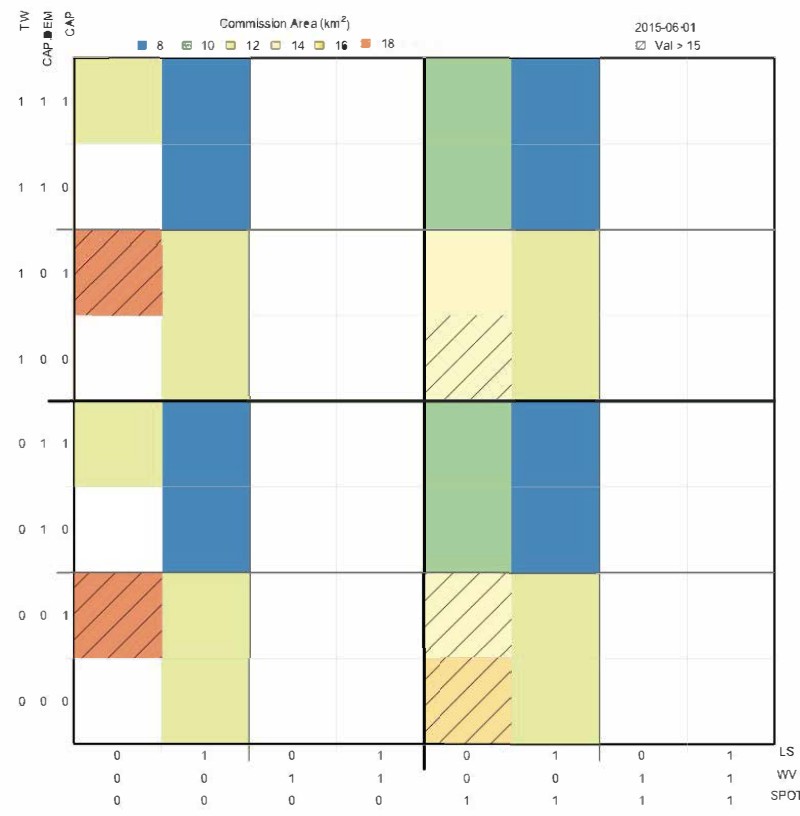

**Figure 13.** Areas of commission for all data combinations available for 1 June 2015.

**Table 2.** Measure of fit, commission, and omission for each day that MOD and HEC-RAS are compared.

| Date | MOP (km$^2$) | HEC-RAS (km$^2$) | Intersect (km$^2$) | Omission (%) | Commission (%) | Fit (%) |
|---|---|---|---|---|---|---|
| 14 April | 7.25 | 7.88 | 4.46 | 27.08 | 29.60 | 43.32 |
| 15 April | 7.25 | 7.66 | 4.44 | 27.59 | 28.83 | 43.58 |
| 16 April | 7.25 | 7.72 | 4.47 | 27.34 | 28.66 | 44.00 |
| 30 April | 6.35 | 8.46 | 4.49 | 19.28 | 34.00 | 46.72 |
| 1 May | 6.35 | 8.11 | 4.49 | 19.47 | 33.48 | 47.05 |
| 2 May | 6.35 | 7.74 | 4.45 | 20.62 | 31.20 | 48.18 |
| 26 May | 13.16 | 11.39 | 6.27 | 40.73 | 22.23 | 37.04 |
| 27 May | 13.21 | 12.97 | 6.70 | 36.19 | 26.62 | 37.19 |
| 28 May | 13.21 | 14.82 | 6.91 | 33.64 | 29.52 | 36.84 |
| 30 May | 22.06 | 22.97 | 12.98 | 31.34 | 23.87 | 44.78 |
| 31 May | 20.97 | 27.35 | 13.95 | 21.95 | 34.43 | 43.61 |
| 1 June | 21.33 | 27.55 | 15.46 | 19.47 | 29.24 | 51.28 |
| 2 June | 17.38 | 26.33 | 13.91 | 12.94 | 35.30 | 51.76 |
| 3 June | 20.13 | 25.21 | 14.76 | 19.79 | 26.84 | 53.37 |

**Table 2.** *Cont.*

| Date | MOP (km$^2$) | HEC-RAS (km$^2$) | Intersect (km$^2$) | Omission (%) | Commission (%) | Fit (%) |
|---|---|---|---|---|---|---|
| 4 June | 9.40 | 22.86 | 7.84 | 7.40 | 55.27 | 37.34 |
| 5 June | 9.17 | 23.15 | 7.02 | 9.99 | 57.54 | 32.47 |
| 6 June | 9.17 | 22.05 | 6.83 | 11.37 | 55.53 | 33.09 |
| 17 June | 15.09 | 15.71 | 7.94 | 34.99 | 26.15 | 38.86 |
| 18 June | 15.09 | 15.71 | 7.94 | 34.99 | 26.15 | 38.86 |
| 19 June | 14.52 | 15.71 | 7.88 | 33.35 | 27.10 | 39.55 |
| Average | | | | | | 42% |

*5.4. Data Comparison: Validation*

Due to the absence of validation data (i.e., FEMA flood extent), YouTube recordings were used to compare the MOP and HEC-RAS outputs. Aerial flood recordings using unmanned aerial vehicles (UAVs) were used as a dataset to visually compare areas of data agreement and disagreement. The videos recorded flood-affected areas throughout the study region and were uploaded around the duration of the 2015 Memorial Day flood. The regions chosen and days used for this comparison were selected based on the YouTube videos and available metadata information provided within the files.

The first day of comparison is during the peak of the event, 1 June 2015, and is located on the southernmost part of the domain area along the Elm Fork Trinity River. Figure 14 shows data agreement of the MOP and HEC-RAS by displaying different data intersections created from a series of data sources. A total of four data sources (LS, CAP, SPOT, and CAP.DEM) are used within this region along with all data combinations available. The HEC-RAS model is overlaid on top of the MOP product along with an image extracted from the YouTube video, located in the top right corner. Observing the output created from the MOP and HEC-RAS, an agreement between the two outputs can be noticed. The image extracted from the UAVs over the same region also shows the area as being flooded.

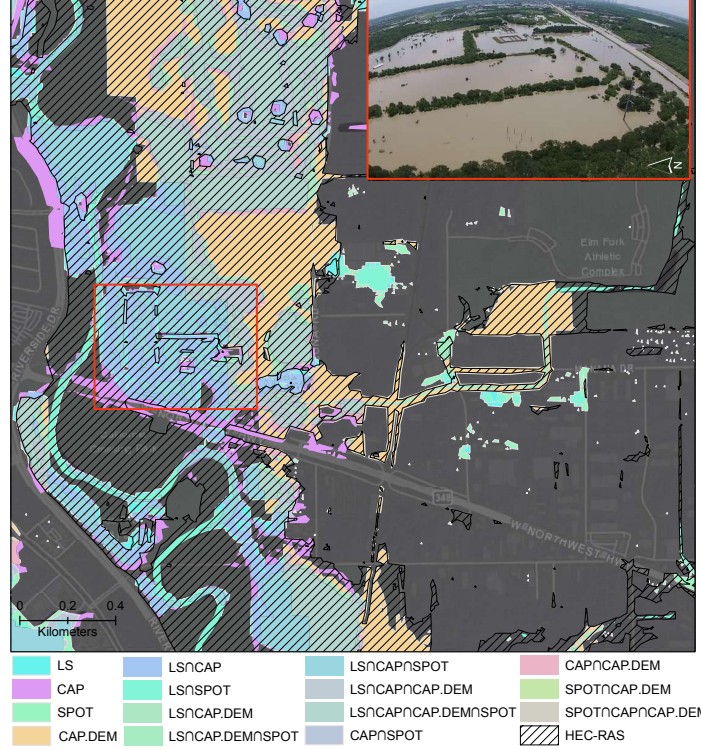

**Figure 14.** Flood inundation for 15 April 2015 estimated by MOP and HEC-RAS.

Figure 15 shows a second example from 3 June 2015 located in the northern part of the study area. In this scenario there is a disagreement between the observation data sources (LS, CAP, CAP.DEM) available and the HEC-RAS output. This is seen by the areas that are not overlapped by the HEC-RAS layer but that are observed as potentially being flooded by the MOP. Looking at the UAV image collected on the same day, it shows that this area is indeed flooded. A possible reason for why this disagreement could occur is because the region shown here is a sports complex made up of multiple baseball fields and the HEC-RAS model may not account for this. This commonly occurs because of data averaging and loss of topographic small-scale details that affect flood propagation.

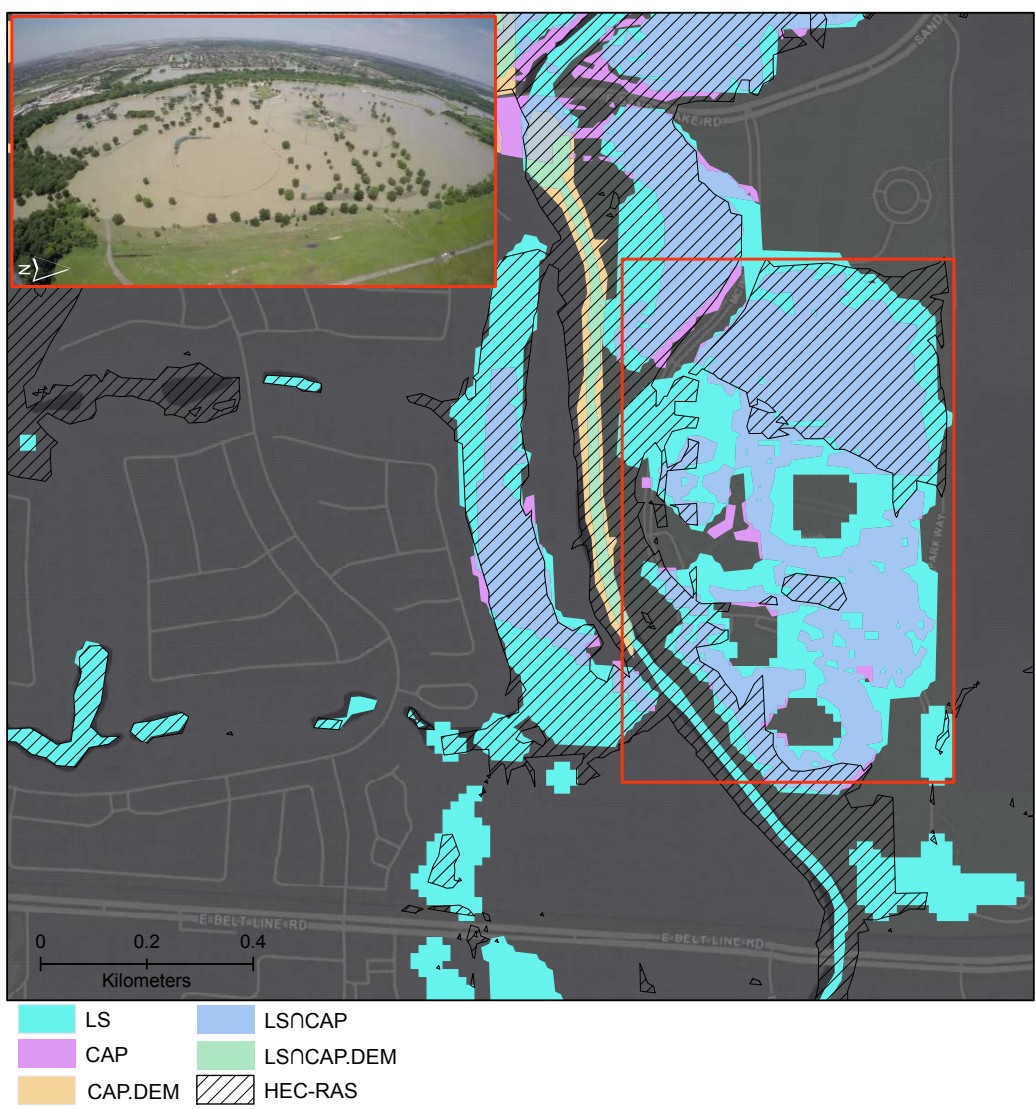

**Figure 15.** Flood inundation for 3 June 2015 estimated by MOP and HEC-RAS. The image in the top left corner shows the condition of the area at the time the UAV recording was collected.

## 6. Conclusions

The recent availability of real-time observations of the Earth and its environment have revolutionized the science of monitoring and understanding disasters. New data- and computation-driven solutions are being applied to study disasters with the common goal to protect people, properties, and the environment. A significant amount of research has been carried out towards creating inundation maps, and often they are created using numerical modeling, remotely sensed data, or a combination of both. These approaches require advanced methodologies, high-performance computing, and cyberinfrastructure

to generate new knowledge from multiple heterogeneous data sources. The proposed domain-independent methodology is applied to solve a long-standing scientific problem of large societal importance: the generation of near-real-time flood inundation maps.

The progression of flood events and data available to observe them change significantly over the duration of an event. Therefore, is it essential to create sequences of high temporal and spatial resolution inundation maps to monitor such changes. Results show that water surfaces estimated by MOP are comparable with simulated output. Contributed data, although they may be geographically sparse, become an important data source when fused with other observational data.

When high-resolution maps are generated using data that vary spatially and temporally, it is important to understand the different contributions that each dataset offers. As shown by Figure 8, for this case study, on average, when a specific data layer is turned off, the days that are covered from that data source also have to be removed. However, throughout different flood phases, the uniqueness of each data source may increase severely, as seen by the CAP, WV, and CAP.DEM. Additionally, there are circumstances in which a data source could also lose its uniqueness if several datasets with similar characteristics are available on the same day. This is shown in Figure 8 when looking at WV and TW. The uniqueness of the data can play a large role with regard to the increase or decrease of the areas of omission and commission.

The temporal component is another factor that needs to be considered. This is based on the number of days that are added or removed with each data layer, but also the water surfaces that are unique to a single data source. This can be best observed when comparing the WV and CAP.DEM layers with the SPOT and TW sources. For this, the WV and CAP.DEM layers contribute the most omission error in comparison to the other datasets as they contain water surfaces that are not observed by any other data layer. This means that as WV and CAP.DEM are removed, there are fewer water features that are predicted by the model. In contrast to the WV and CAP.DEM sources, on average, SPOT and TW provide very little information to the MOP. This is because for the days that SPOT and TW data are available, there are several other data sources that are fused for those days; thus, the other sources help augment the absence of SPOT and TW but not for the WV and CAP.DEM data.

A limitation of this product is that currently it is generated only with data that are dependent on only daytime observations. The satellite observations used are limited by the passive sensor's ability to collect data only during daytime hours. Therefore, the MOP does not have the ability to map nighttime inundation. Because of the daytime constraints associated with the remote sensing data, other available data, such as social media, must be tailored and fused with alternative data that may be available in this timeframe. Nighttime social media data can be used; however, the level of uncertainty is greater as the information provided could be obscured by nighttime conditions.

A multiscale observation product (MOP) was created with the capability to dynamically integrate various data sources to estimate flood inundation and assess damage during an event. It aims to provide scientists in the field with a suitable method of integrating heterogeneous remote sensing data with social media photos and to fill the need for an independent product that can be used for model parameterization, validation, and assimilation. While in the current study, the 2D MOP maps are created by aggregating all data available in a single day, the assessment may be performed at a finer temporal resolution. Lastly, although this method was applied to a flood event, the same methodology can be applied to different hazard-related studies and emergencies.

**Author Contributions:** Conceptualization, E.S.; Software, G.C., and E.S.; Investigation, E.S., and A.K.; Writing—original draft, E.S.; Writing—review & editing, E.S., and G.C. All authors have read and agreed to the published version of the manuscript.

**Funding:** This work was partially supported by the Office of Naval Research (ONR) award no. N00014-16-1-2543 (PSU no. 171570).

**Data Availability Statement:** The multispectral data presented in this study are openly available for download through the U.S. Geological Survey Hazards Data Distribution System (http://hdds.usgs.gov) The web portal is intentionally designed to assist experts in the course of emergencies by providing imagery and documents acquired before, during, and after an event. Additional satellite-derived products (e.g., DEM, land cover) and tools can be downloaded using the USGS EarthExplorer tool (https://earthexplorer.usgs.gov/) by querying for desirable datasets. Other data used in this study were obtained through open API or through active collaborations built with universities and research centers.

**Acknowledgments:** We wish to thank Alexander Klippel, Anthony Robinson, and Andrea Tapia for their comments and recommendations that helped improve the present manuscript.

**Conflicts of Interest:** The authors declare no conflict of interest.

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
