# Peer review of "Multiscale Observation Product (MOP) for Temporal Flood Inundation Mapping of the 2015 Dallas Texas Flood"

_remotesensing, doi:10.3390/rs15061615_

Round 1

Reviewer 1 Report

The manuscript describes a method to combine different kind of instrumental observations and proxy data for improving flood inundation maps during emergencies. The method used is very interesting because uses open-source data and the contribution of normal people that can contribute in real time to data collection with photos and observations. The manuscript is well written, describes extensively all the source data used, and examines in detail the results obtained. I will suggest to public it with some minor revisions that need to clarify, in my opinion, some questions:

In section 4.1 a methodology is described to join multi-scale observations and in line 308 a “weighted sum” is named. It could be interesting specify in the text or in a table the nature and the value of the weighs.

In section 4.6 authors affirm that they re-sample data to uniform the spatial resolution of the multiple sources. Probably they assumed that the procedure is known, but, in my opinion, a brief description or the name of the method used is necessary.

On line 498: bathy-metry, probably authors intend bathymetry.

Author Response

Added in the attached file

Author Response

Added in the attached file
